# Colchicine Use and Major Adverse Cardiovascular Events in Male Patients with Gout and Established Coronary Artery Disease: A Veterans Affairs Nested Retrospective Cohort Study

**Gary H. Ho** [1,2,*,†] , **Michael Toprover** [1,2,†] , **Daria B. Crittenden** [1,2] , **Binita Shah** [3,4,‡] **and Michael H. Pillinger** [1,2,‡]

1   Rheumatology Section, Department of Medicine, Margaret Cochran Corbin Campus of the New York Harbor Health Care System, United States Department of Veterans Affairs, New York, NY 10010, USA
2   Division of Rheumatology, Department of Medicine, New York University Grossman School of Medicine, New York, NY 10016, USA
3   Cardiology Section, Department of Medicine, Margaret Cochran Corbin Campus of the New York Harbor Health Care System, United States Department of Veterans Affairs, New York, NY 10010, USA
4   Division of Cardiology, Department of Medicine, New York University Grossman School of Medicine, New York, NY 10016, USA
*   Correspondence: gary.ho@nyulangone.org
†   These authors contributed equally to this work.
‡   These authors also contributed equally to this work.

**Abstract:** Background: Despite colchicine's proven efficacy in the non-gout population, the effects of colchicine on the risk of major adverse cardiovascular events (MACE) among high-risk patients with gout remain to be determined. The purpose of this study is to evaluate the association between colchicine use and MACE in gout patients with preexisting coronary artery disease (CAD). Methods: This retrospective cohort study followed patients with gout and established CAD within the VA New York Harbor Healthcare System who did or did not use colchicine regularly (>30 continuous days prescription with at least 1 refill). The primary outcome was first MACE, defined as a composite of non-fatal myocardial infarction, coronary artery bypass graft, non-fatal stroke, and all-cause mortality. Part I of the primary analysis compared MACE between colchicine users and nonusers. Part II of the study compared MACE within the colchicine-use group, divided into quartiles based on consistency of colchicine use (i.e., percentage of time on colchicine). Results: Among 1638 patients with gout, 355 had established CAD (239 colchicine users and 116 nonusers). In this cohort, the odds of MACE were similar between any colchicine use compared to nonuse (OR 1.14; 95% CI (0.59–2.20)); however, colchicine users overall had a higher baseline cardiovascular risk profile than nonusers, suggesting that colchicine may have served to equilibrate risk between the two groups. Moreover, patients in the highest continuous colchicine-use quartile (>70% of observation period on colchicine) demonstrated lower odds of MACE compared to those in the lowest three quartiles (OR 0.35; 95% CI (0.13–0.93)), with no difference in baseline risk. Additionally, colchicine users had a numerically lower rate of MACE during periods of active use compared with periods of lapse. Kaplan–Meier analysis revealed a difference in cumulative MACE over time, favoring the subgroup with the most consistent colchicine use ($p_{log-rank}$ = 0.01). Conclusions: Despite higher CV risk, gout patients with CAD receiving colchicine had no higher rates of MACE than those not receiving colchicine. Among all patients with gout and CAD treated with colchicine, those with the most consistent colchicine use had lower odds of MACE, and event rates were lower during active use. Colchicine protection against cardiovascular events may require maintenance of colchicine bloodstream levels.

**Keywords:** gout; colchicine; cardiovascular risk; major adverse cardiovascular events



## 1. Introduction

Gout is the most common inflammatory arthritis, affecting approximately 3.9% of the U.S. adult population [1]. Patients with gout are more likely to have cardiovascular (CV) risk

factors and comorbidities than those without gout, including renal disease, hypertension, metabolic syndrome, and diabetes mellitus [2]. In addition, patients with gout experience chronic, low-level inflammation between flares, potentially increasing CV risk beyond their associated traditional CV risk factors [3–5]. Therefore, the overall CV risk associated with the presence of gout may be underestimated and underrecognized.

Colchicine, a foundational medication for treating and preventing gout flares [6], has recently been tested in randomized placebo-controlled trials of all-comers with established coronary artery disease (CAD) and was shown to be effective in reducing major adverse cardiovascular events (MACE). The Colchicine Cardiovascular Outcomes Trial (COLCOT) enrolled patients with recent myocardial infarction (MI) and found that 0.5 mg of colchicine daily reduced MACE by 23% relative to placebo [7]. Similarly, the Low Dose Colchicine 2 (LoDoCo2) trial enrolled patients with established chronic CAD and demonstrated that colchicine 0.5 mg daily resulted in a relative risk reduction of MACE by 29% compared to placebo [8]. Based on these data, colchicine was recently approved in Canada for the reduction of MACE risk in stable ischemic heart disease [9]. However, both studies enrolled only limited numbers of patients with gout and excluded those already taking long-term colchicine; therefore, these findings may not extend to the gout population. When considering its potential for long-term use, low-dose colchicine has the added benefits of an excellent safety profile, relative affordability, and long-term tolerability [10–12].

The mechanisms by which colchicine may reduce MACE are incompletely defined. Colchicine inhibits microtubules and has many targets vis-à-vis leukocyte activation and adherence, thereby reducing leukocyte–endothelial adhesion and leukocyte–platelet aggregation [13–16]. Colchicine also inhibits IL-1β production, and a biologic agent (canakinumab) that targets IL-1β has been shown to reduce MACE in patients with a prior MI and elevated C-reactive protein [17]. When administered prior to percutaneous coronary intervention (PCI), colchicine attenuated a rise in inflammatory markers and reduced periprocedural myocardial injury [18,19]. One hypothesis for the cardioprotective effect of colchicine is that it may stabilize plaque during acute rupture or erosion, minimizing downstream thrombosis and end-organ ischemia.

The cardiovascular benefits of colchicine in gout are less well documented. Crittenden et al. performed a cross-sectional analysis and reported an association between colchicine use and a reduced MI rate in a male gout population (54% reduction among colchicine users versus placebo) [20]. Subsequently, Solomon et al. recapitulated these findings utilizing Medicare claims data, concluding that colchicine use in gout was associated with a 49% reduction in CV events and a 73% reduction in all-cause mortality compared to those not prescribed colchicine [21]. Most recently, we employed a nested cohort from among a longitudinal retrospective cohort of gout patients in the Veterans Affairs New York Harbor Health Care System (NYHHCS) and demonstrated that in patients without preexisting CAD, the use of colchicine was associated with a lower risk of incident CAD in the absence of chronic kidney disease (CKD) [22].

In contrast to the aforementioned studies, the effects of colchicine on the risk of MACE among patients with gout and established CAD remain to be determined. The American College of Rheumatology (ACR) guidelines for the management of gout focus expressly on colchicine use to treat or prevent gout flares for a duration of only 3–6 months [6]. Moreover, ACR guidelines do not distinguish between colchicine, non-steroidal anti-inflammatory drugs (NSAIDs), and corticosteroids for flare prophylaxis, despite the well-established associations with CV risk of the latter two agents [23,24]. It is therefore crucial to determine whether the reported beneficial effects of colchicine on MACE are sustained in patients with gout—itself a potent CV risk factor—and already established CAD, or if this subgroup of patients is already at too high a risk to benefit from colchicine. To address this needs gap, we assessed the association between colchicine use and MACE among a longitudinal, nested cohort of patients with gout and established CAD. We hypothesized that colchicine use would be associated with reduced MACE and that more consistent colchicine use would result in a larger CV protective benefit.

## 2. Materials and Methods

The cohort from which our study group was derived has been previously described [22]. Briefly, all patients with ICD-9 codes for gout and/or hyperuricemia with a clinical visit to the VA NYHHCS between 1 January 2000 and 31 December 2009 were identified (chosen specifically to encompass a period prior to the release of the 2012 ACR Gout Treatment Guidelines, after which extended colchicine use, beyond the period around the initiation of urate lowering, was not recommended). Additionally, patients were required to meet at least one of the following criteria for gout diagnosis: presence of monosodium urate crystals on microscopy (synovial fluid or tophus); ≥6/12 of the 1977 American Rheumatology Association (ARA) gout classification criteria; ≥4/12 ARA criteria (allopurinol permitted as a surrogate for hyperuricemia) plus a primary care physician diagnosis of gout; or a diagnosis of gout by a rheumatologist. Patients were excluded if they were younger than 45 or older than 90 years of age. Female patients were excluded since they constituted only 0.6% of the potential population. Participants with a late index date for gout diagnosis (≤3 months prior to the end of the study observation period) were excluded due to an inadequate opportunity for follow-up. From among the resulting population, participants with preexisting CAD were identified based on an electronic medical record (EMR) problem list and confirmed by direct chart review. CAD was defined as any of the following: documentation of CAD by a primary care physician or cardiologist; positive cardiac stress test; ≥50% stenosis on coronary angiography; prior MI; and prior coronary revascularization with coronary artery bypass graft (CABG) or PCI. Participants without preexisting CAD were excluded.

Data on colchicine use was obtained through the VA pharmacy database, which captures all prescriptions written by VA providers. We defined colchicine users as those who filled an initial prescription(s) for colchicine for at least one period of >30 consecutive days and refilled their prescription at least once. Nonusers were defined as never having filled a prescription for colchicine throughout the study duration. Patients filling prescriptions for ≤30 consecutive days were considered acute-only users (e.g., for gout flare treatment alone) and were excluded. Within the "colchicine-use" group, participants were further divided into quartiles based on "consistency" of colchicine use, defined as the total days of colchicine dispensed over the duration of the observation period for the individual patient (i.e., percentage of time on colchicine).

For each participant in the colchicine-use group, the observation period started on the first day colchicine was dispensed, and for the "nonuser" group, observation started on the first clinical visit for gout in the EMR within the study period. The observation period extended from the participant's enrollment date to December 31st, 2009, or the last clinic visit, whichever occurred first.

Each participant's EMR was reviewed for demographics, baseline characteristics (including comorbidities and other risk factors for CV disease), medication use, anthropometric data, and laboratory tests at the time point closest to study enrollment. Direct chart review was used to identify events during the observation period. The primary outcome was first-observed MACE, defined as non-fatal MI, CABG, non-fatal stroke, or all-cause mortality. Our definition of MI included both definite and probable MIs. Definite MI was defined as a rise in troponin or CK–MB (creatine kinase–myocardial band) accompanied by a clinical note confirming MI, regardless of treatment. Probable MI was defined as physician documentation of MI without available measurements of troponin or CK–MB, most commonly representing an MI event occurring outside of the VA health system (VA records outside of the NYCHHCS were also reviewed). Stroke (CVA) was defined as a neurologist-documented ischemic stroke, evidenced by an episode of neurologic dysfunction lasting more than 24 h with compatible findings on brain imaging. Secondary outcomes included the individual components of MACE.

Part I of the study's primary analysis compared MACE between colchicine users and nonusers. Part II of the study examined only colchicine users and compared MACE

between colchicine users with the most consistent use (highest colchicine-use quartile) and those with inconsistent use (lowest three quartiles).

Baseline variables between groups were compared using Fisher's exact test for categorical variables and a 2-sample independent *t*-test for normally distributed variables, or the Mann–Whitney U test for skewed continuous variables. Primary end-point analyses utilized Fisher's exact test, logistic regression, and Kaplan–Meier survival analysis. Reported incidences were expressed per observation-year to account for variations in the length of subject follow-up. Logistic regression analyses were adjusted for covariables: age, hypertension, diabetes mellitus, dyslipidemia, CKD, prior MI, allopurinol use, NSAID use, and statin use. Secondary end-points were analyzed in the same manner as the primary outcome. Significance was set at a 2-sided alpha level of 0.05, and analyses were performed using SPSS version 25 (IBM, Armonk, NY, USA).

This study was approved by the Institutional Review Board of the VA New York Harbor Health Care System.

## 3. Results

### 3.1. Patient Identification

178,877 patients had active medical records within the VA NY Harbor Health Care System between 2000 and 2009. Furthermore, 7819 patients had ICD-9 codes listed for gout and/or hyperuricemia, indicating potential gout. Of these, 1638 patients met our more stringent study criteria for clinical gout by chart review. Broken down by gout-specific inclusion criteria, 130 met inclusion by a crystal diagnosis, 681 were rheumatologist-diagnosed, 223 met $\geq 6/12$ ARA classification criteria, and 604 had $\geq 4/12$ ARA criteria along with a diagnosis of gout by their primary care physician. After further inclusion and exclusion criteria were applied (Figure 1), 239 colchicine users and 116 nonusers with preexisting coronary artery disease were identified (*n* = 335).

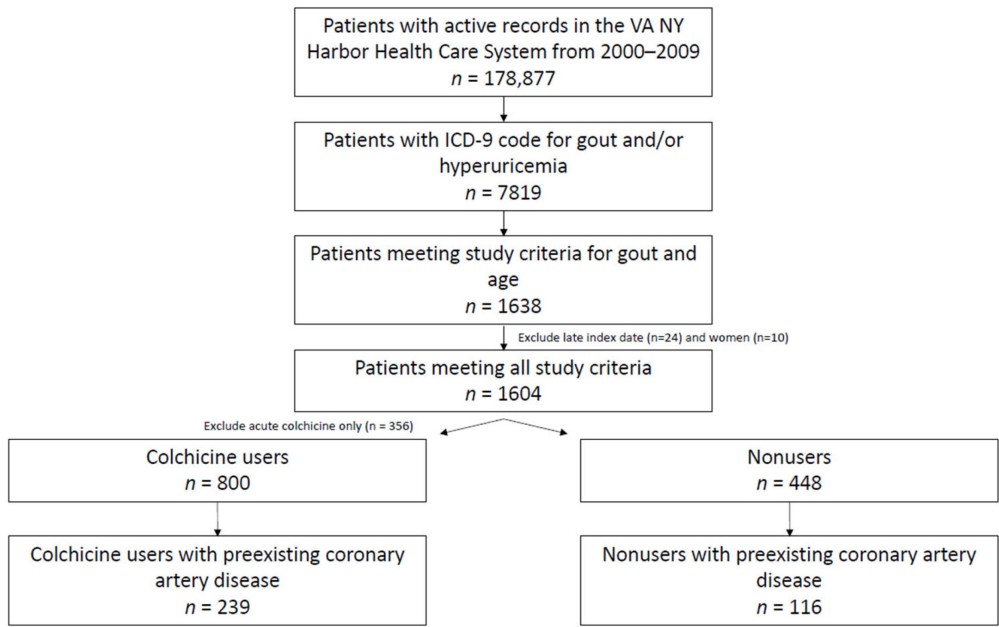

**Figure 1.** Study population. Flow chart of study population selection.

### 3.2. Part I: Colchicine Nonusers versus Users

Demographics of colchicine users and nonusers are shown in Table 1. Both the colchicine user and nonuser groups had elevated BMIs (30.8 kg/m$^2$ vs. 31.4 kg/m$^2$) and a strong history of prior tobacco use (81.2% vs. 81.0%). A similar proportion of patients in both groups had hypertension, diabetes mellitus, and CKD, but colchicine users had higher LDL-cholesterol levels (102 ± 37 mg/dL vs. 91 ± 31 mg/dL; *p* = 0.03) and a more

common prior history of PCI (34.3% vs. 22.4%; $p$ = 0.03) and MI (45.6% vs. 34.5%; $p$ = 0.04), suggesting more severe coronary disease and, potentially, a higher risk for future CV events. More participants in the colchicine group were prescribed allopurinol (27.2% vs. 16.4%) and NSAIDs (30.1% vs. 19.0%), suggesting differences in either gout severity or management. On average, the observation period in colchicine users was longer than in nonusers (61 ± 36 months vs. 48 ± 36 months; $p \leq 0.01$). Of note, dosing prescriptions for colchicine varied by patient as follows: 58.8% of colchicine users were prescribed 0.6 mg once daily, 13% 0.6 mg twice daily, 9.7% less than 0.6 mg daily, and the rest received a combination of daily and BID scripts during their observation periods (18.5%).

**Table 1.** Baseline characteristics of colchicine users versus nonusers.

| Variable | Nonusers ($n$ = 116) | Colchicine Users ($n$ = 239) | $p$ Value |
|---|---|---|---|
| Age, years ± SD | 77 ± 10 | 78 ± 9 | 0.24 |
| Race | | | 0.16 |
| White | 64.0% | 55.6% | |
| Black | 26.1% | 34.9% | |
| Asian-Pacific Islander | 4.5% | 2.6% | |
| Ethnicity | | | 0.23 |
| Not Hispanic | 91.9% | 84.3% | |
| Hispanic | 6.3% | 10.6% | |
| Body mass index, kg/m$^2$ ± SD | 31.4 ± 5.1 | 30.8 ± 4.3 | 0.62 |
| Diabetes mellitus | 37.1% | 42.7% | 0.30 |
| Hypertension | 89.7% | 87.9% | 0.58 |
| Systolic blood pressure, mmHg ± SD | 139 ± 21 | 131 ± 21 | <0.01 |
| Diastolic blood pressure, mmHg ± SD | 74 ± 13 | 74 ± 13 | 1.00 |
| Dyslipidemia | 65.5% | 76.2% | 0.07 |
| LDL cholesterol, mg/dL ± SD | 91 ± 31 | 102 ± 37 | 0.03 |
| Prior myocardial infarction | 34.5% | 45.6% | 0.04 |
| Prior coronary artery bypass graft | 31.0% | 28.0% | 0.62 |
| Prior percutaneous coronary intervention | 22.4% | 34.3% | 0.03 |
| Chronic kidney disease | 42.2% | 42.3% | 0.91 |
| Serum creatinine, mg/dL ± SD | 1.6 ± 1.2 | 1.6 ± 1.1 | 1.00 |
| Glomerular filtration rate, mL/min ± SD | 59 ± 21 | 54 ± 22 | 0.05 |
| History of tobacco use | 81.0% | 81.2% | 1.00 |
| Current tobacco use | 18.1% | 14.6% | 0.44 |
| Allopurinol use | 16.4% | 27.2% | 0.03 |
| Serum uric acid, mg/dL ± SD | 8.8 ± 2.4 | 8.7 ± 2.1 | 0.79 |
| NSAID use | 19.0% | 30.1% | 0.03 |
| Aspirin use | 50.0% | 59.8% | 0.11 |
| Statin use | 62.9% | 64.9% | 0.81 |
| ACE inhibitor use | 62.9% | 52.3% | 0.05 |
| β-blocker use | 62.9% | 71.1% | 0.18 |
| Any antihypertensive use | 69.8% | 66.1% | 0.47 |
| Mean daily colchicine dose, mg ± SD | 0.0 ± 0.0 | 0.71 ± 0.25 | – |
| Time on Colchicine, % of observation period ± SD | 0 ± 0 | 46.8 ± 29.8 | – |

Continuous data are shown as mean ± standard deviation and compared using the 2-sample independent *t*-test for normally distributed, or Mann–Whitney U test for skewed continuous variables. Categorical data are shown as a percentage of the total subgroup and compared using the Fisher's exact test. SD, standard deviation; LDL, low-density lipoprotein; NSAID, nonsteroidal anti-inflammatory drug, ACE, angiotensin-converting enzyme.

Despite a higher cardiovascular risk profile (defined as an increased prevalence of prior MI, prior PCI, NSAID use, and a higher baseline LDL cholesterol) in the colchicine users group, the incidence of MACE per year of observation was similar among users and nonusers (2.9% vs. 3.7%; *p* = 0.38), with a trend towards decreased MACE in the colchicine use group (Figure 2A, Supplementary Table S2). The odds ratio of experiencing a MACE in the colchicine use group and nonuse groups was also not significantly different (OR 1.28; 95% CI, 0.69–2.35; *p* = 0.43). Adjustment for covariables yielded a further attenuation of the odds ratio of MACE in colchicine users compared to nonusers (OR 1.14; 95% CI, 0.59–2.20; *p* = 0.70). Kaplan–Meier survival analysis revealed no significant difference in time-to-first major adverse cardiovascular events (*p* = 0.99) (Figure 2B).

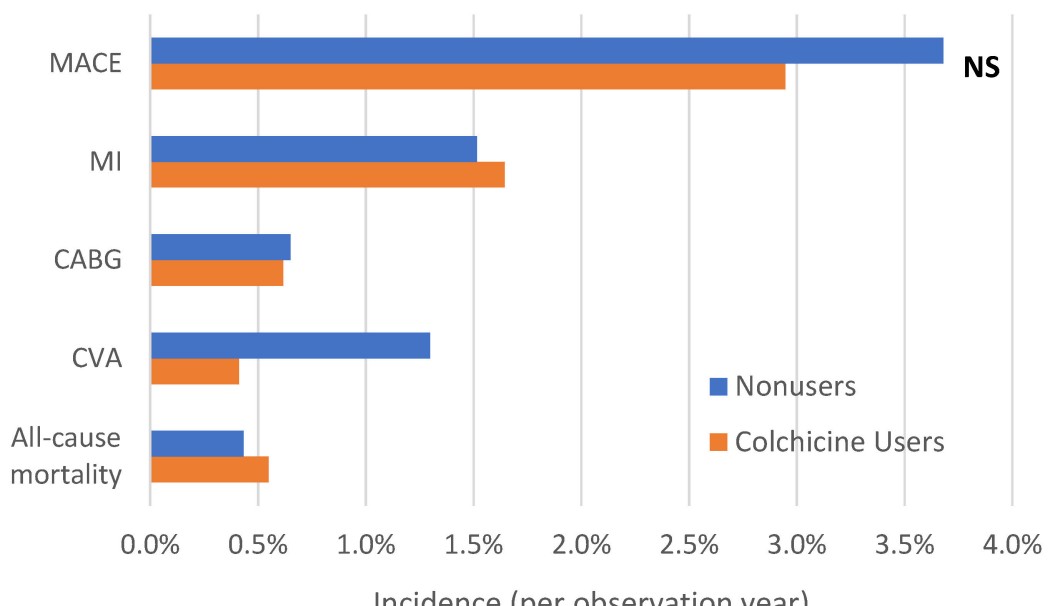

(**A**)

**Figure 2.** *Cont.*

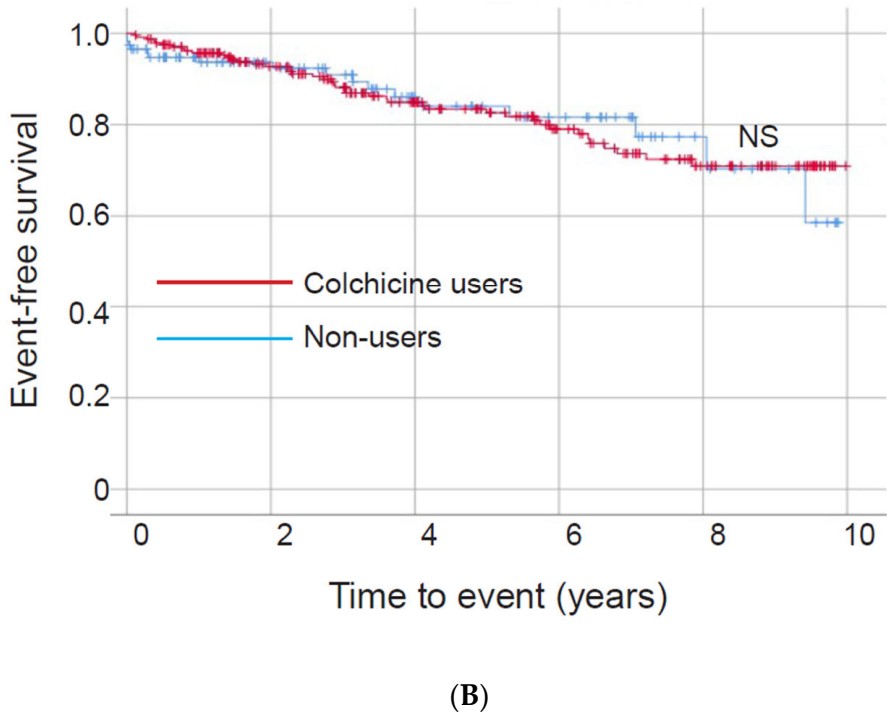

**(B)**

**Figure 2.** Major adverse cardiovascular events in colchicine users and nonusers. (**A**) Incidence of MACE and component events, reported per observation year, in colchicine users and nonusers. For MACE, only the first event was counted if patients experienced more than one component event during the observation period. The difference in incidence of CVA between nonusers and colchicine users was statistically significant ($p = 0.048$). (**B**) Kaplan–Meier survival curves in colchicine users and nonusers illustrating the cumulative proportion of patients experiencing a MACE over time; maximum observation period of 10 years. Patients censored at the end of their respective observation periods are demarcated by a vertical line. Log rank (Mantel–Cox) $p = 0.99$. NS = non-significant.

### 3.3. Part II: Colchicine Group—Consistency of Colchicine Use

Among colchicine users, the majority spent a limited amount of the observation period actually taking colchicine (Figure 3A), which potentially contributed to the lack of a MACE difference between colchicine users and nonusers. For the second part of our study, colchicine users were therefore divided into four equal quartiles based on the calculated consistency of colchicine use (percentage of time on colchicine): Quartile 1 (11.4% ± 4.9%, $n = 60$), quartile 2 (31.3% ± 7.4%, $n = 60$), quartile 3 (56.6% ± 7.8%, $n = 60$), and quartile 4 (88.5% ± 9.9%, $n = 59$). The mean and standard deviation for each reported quartile group were calculated by averaging the total days of colchicine dispensed over the duration of the observation period for each individual patient. Overall, patients in quartile 4 were prescribed colchicine for the large majority (mean 88.5%) of their observation periods. Because patients from quartiles 1–3 collectively received colchicine less than 50% of the time, patients from these quartiles were combined and compared with Quartile 4. Baseline characteristics were similar between the two groups (Table 2) and between all quartiles (Supplementary Table S1), suggesting comparable CV risk.

The annualized incidence of MACE was 1.5% in quartile 4, compared with 3.4% in quartiles 1–3 ($p = 0.01$). The odds ratio for MACE in quartile 4 versus quartiles 1–3 was 0.35 (95% CI, 0.13–0.93; $p = 0.03$) after adjusting for all covariables. Incidences of MI, CABG, CVA, and all-cause mortality individually were numerically lower among subjects with more consistent colchicine use (Figure 3B, Supplementary Table S3). Kaplan–Meier analysis revealed a significant difference in cumulative MACE over time, favoring colchicine users in quartile 4 over quartiles 1–3 ($p = 0.01$) (Figure 3C).

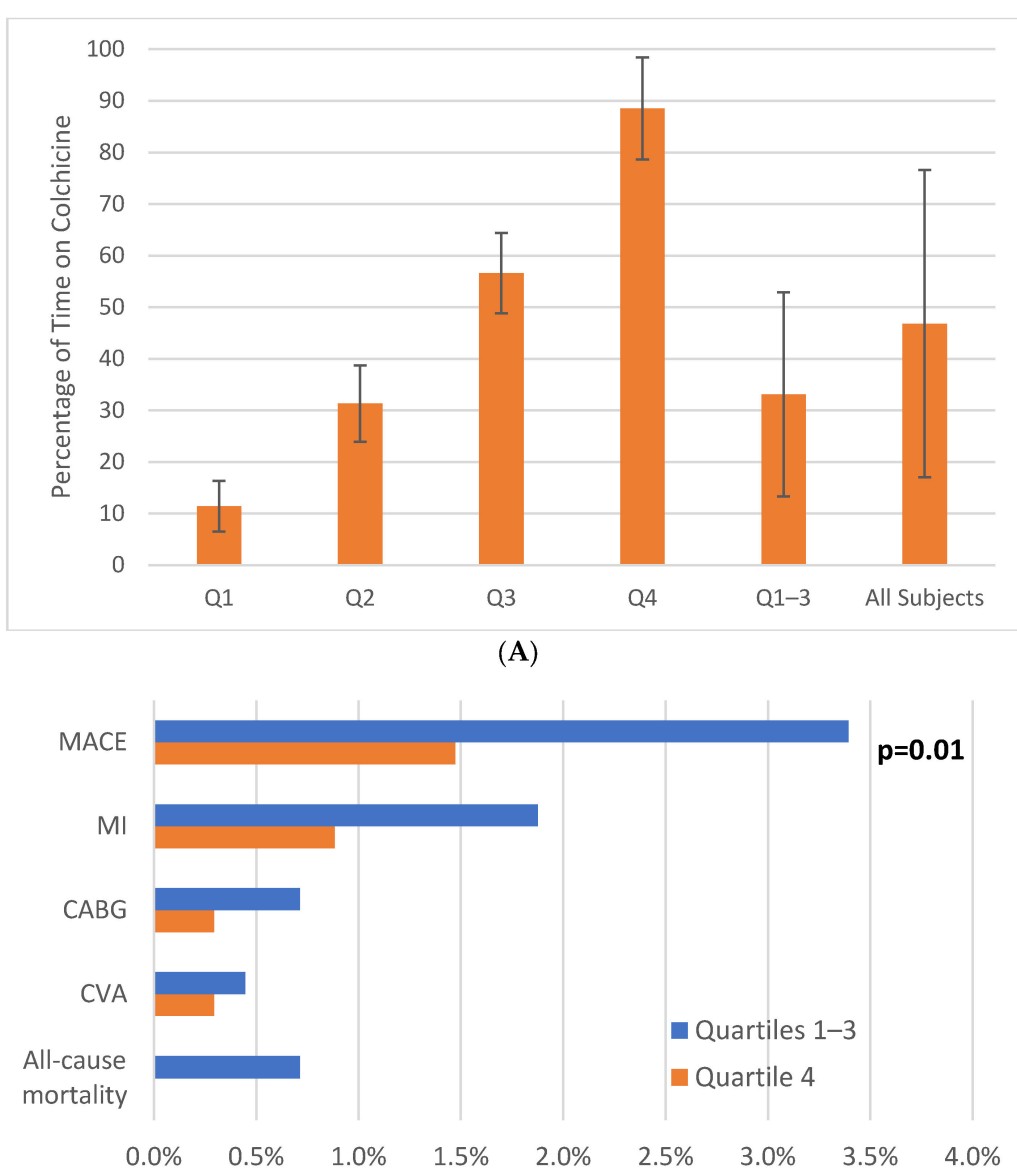

**Figure 3.** *Cont*.

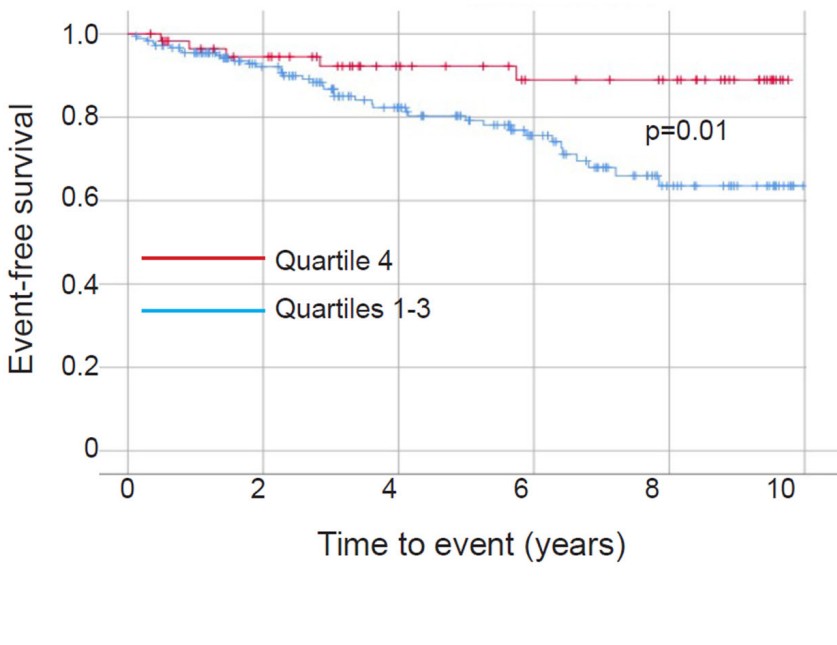

(**C**)

**Figure 3.** Quartiles of colchicine use and major cardiovascular events in consistent and inconsistent colchicine users. (**A**) Mean percentage of time on colchicine within each quartile, quartiles 1–3, and all subjects. Vertical bars represent the standard deviation. (**B**) Incidence of MACE and component events, reported per observation year, in quartiles 1–3 and quartile 4. For MACE, only the first event was counted if patients experienced more than one component event during the observation period. The difference in incidence of all-cause mortality between Q4 and Q1–3 was statistically significant ($p$ = 0.02). (**C**) Kaplan–Meier survival curves in quartiles 1–3 and quartile 4 illustrating the cumulative proportion of patients experiencing a MACE over time; maximum observation period of 10 years. Patients censored at the end of their respective observation periods are demarcated by a vertical line.

**Table 2.** Baseline characteristics of combined quartiles 1–3 versus quartile 4.

| Variable | Quartiles 1–3 ($n$ = 180) | Quartile 4 ($n$ = 59) | $p$ Value |
|---|---|---|---|
| Age, years $\pm$ SD | 78 $\pm$ 10 | 76 $\pm$ 9 | 0.17 |
| Race | | | 0.93 |
| White | 54.6% | 58.6% | |
| Black | 33.9% | 37.9% | |
| Asian-Pacific Islander | 2.3% | 3.4% | |
| Ethnicity | | | 0.81 |
| Not Hispanic | 83.1% | 88.1% | |
| Hispanic | 10.2% | 11.9% | |
| Body mass index, kg/m$^2$ $\pm$ SD | 30.9 $\pm$ 5.8 | 31.0 $\pm$ 5.1 | 0.92 |
| Diabetes mellitus | 43.9% | 39.0% | 0.55 |
| Hypertension | 88.9% | 84.7% | 0.34 |
| Systolic blood pressure, mmHg $\pm$ SD | 132 $\pm$ 22 | 129 $\pm$ 20 | 0.43 |
| Diastolic blood pressure, mmHg $\pm$ SD | 74 $\pm$ 13 | 73 $\pm$ 12 | 0.40 |

**Table 2.** *Cont.*

| Variable | Quartiles 1–3 (*n* = 180) | Quartile 4 (*n* = 59) | *p* Value |
|---|---|---|---|
| Dyslipidemia | 74.4% | 81.4% | 0.38 |
| LDL cholesterol, mg/dL ± SD | 104 ± 40 | 94 ± 26 | 0.37 |
| Prior myocardial infarction | 44.4% | 49.2% | 0.55 |
| Prior coronary artery bypass graft | 29.4% | 23.7% | 0.50 |
| Prior percutaneous coronary intervention | 33.9% | 35.6% | 0.75 |
| Chronic kidney disease | 41.7% | 44.1% | 0.76 |
| Serum creatinine, mg/dL ± SD | 1.6 ± 1.3 | 1.4 ± 0.5 | 0.88 |
| Glomerular filtration rate, mL/min ± SD | 53 ± 24 | 55 ± 18 | 0.76 |
| History of tobacco use | 81.1% | 81.4% | 1.00 |
| Current tobacco use | 13.3% | 18.6% | 0.40 |
| Allopurinol use | 30.0% | 18.6% | 0.09 |
| Serum uric acid, mg/dL ± SD | 8.6 ± 2.1 | 8.9 ± 2.0 | 0.50 |
| NSAID use | 32.2% | 23.7% | 0.25 |
| Aspirin use | 58.9% | 62.7% | 0.65 |
| Statin use | 61.7% | 74.6% | 0.08 |
| ACE inhibitor use | 51.1% | 55.9% | 0.55 |
| β-blocker use | 70.9% | 72.9% | 0.90 |
| Any antihypertensive use | 65.0% | 69.5% | 0.64 |
| Mean daily colchicine dose, mg ± SD | 0.72 ± 0.26 | 0.69 ± 0.22 | 0.43 |
| Time on colchicine, % of observation period ± SD | 33.1 ± 19.8 | 88.5 ± 9.9 | <0.01 |

Continuous data are shown as mean ± standard deviation and compared using the 2-sample independent *t*-test for normally distributed, or Mann–Whitney U test for skewed continuous variables. Categorical data are shown as a percentage of the total subgroup and compared using the Fisher's exact test. SD, standard deviation; LDL, low-density lipoprotein; NSAID, nonsteroidal anti-inflammatory drug, ACE, angiotensin-converting enzyme.

### 3.4. Exploratory Outcome: Major Adverse Cardiovascular Events during Active Colchicine Use versus Lapse

We next assessed whether any possible CV benefit of colchicine required its active use. CV events within the colchicine-use group overall were adjudicated as to whether participants were taking colchicine at the time of the event. A CV event was categorized as occurring during "active use" if the event took place during a period covered by pharmacy-dispensed colchicine, including the 14 days after a refill was expected to run out. An event was categorized as occurring during a "lapse period" if it occurred >14 days after a prescription ran out (to allow time for drug elimination, and account for residual pills that patients may have continued taking beyond the prescription period). Lapses were considered to persist until the next date of a prescription refill or re-prescription. There were numerically more MACE, MIs, CABGs, and higher all-cause mortality during lapse periods compared to periods of active use (Figure 4).

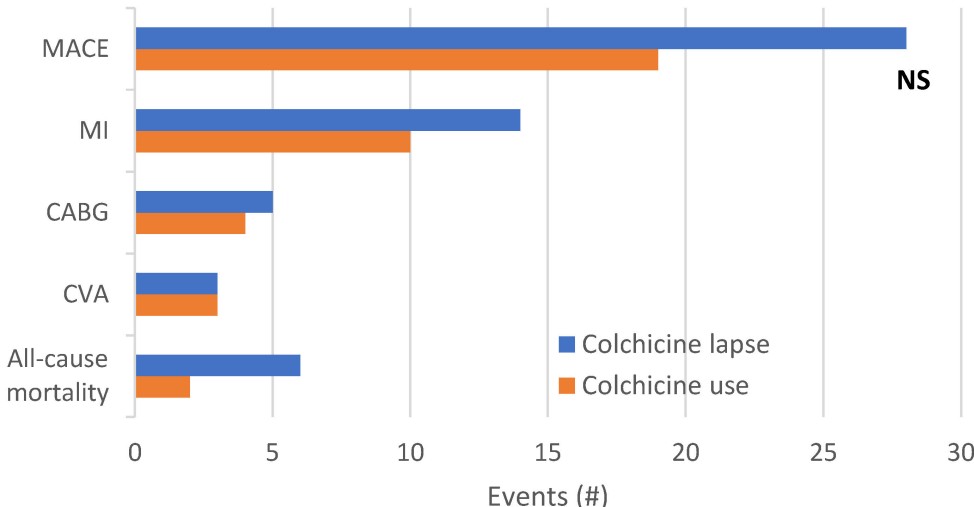

**Figure 4.** Major adverse cardiovascular events during colchicine use and lapse. Total number of major adverse cardiovascular events occurring during active colchicine use and lapse periods. Events were compared using the Fisher's exact test. NS = non-significant.

## 4. Discussion

In this cohort of VA gout patients with preexisting coronary artery disease, we initially compared MACE among patients with gout on or off chronic colchicine. Compared with the gout-limited LoDoCo2 trial, our overall population was older in age, had more comorbid hypertension, diabetes, and CKD, and substantially less statin use despite known CAD [8]. Although the gout patients prescribed colchicine had a higher baseline level of CV risk, we observed no significant difference in MACE outcomes between the colchicine users and nonusers. Because the duration and number of CV comorbidities may limit the efficacy of colchicine on CV risk [25], adjusted analyses were performed. After adjusting for covariables, the odds ratio of a major adverse cardiovascular event with colchicine use further corrected towards no difference in outcomes, supporting the hypothesis that colchicine use may have served to mitigate CV risk in the users group to a level equivalent to that in the nonusers group.

The percentage of time that colchicine users spent on colchicine varied widely, and most users spent only a minority of their observation periods on colchicine, suggesting that whole-group comparisons might not actually capture the full impact of colchicine use. The second portion of our study therefore separated the colchicine users group into quartiles according to the consistency of use and compared quartiles 1–3 collectively with quartile 4. In these analyses, there was no difference in baseline CV risk between quartiles 1–3 and quartile 4. We observed that patients with the highest percentages of time on colchicine (quartile 4) experienced a lower incidence of MACE compared to quartiles 1–3. The odds ratio of experiencing a major adverse cardiovascular event was considerably lower in the high-use group, suggesting that consistent colchicine use in gout patients provided CV protective effects.

To account for the confounder of observation time on the primary outcome, Kaplan–Meier survival analyses were employed. In the user/nonuser comparisons, cumulative incidence curves were similar between colchicine users and nonusers. In contrast, among the colchicine users, the subgroup with the most consistent colchicine use (quartile 4) was significantly less likely to experience MACE over the study period. The Kaplan–Meier survival curves for high and low use separated around the 2-year mark and continued to diverge (Figure 3C), suggesting that colchicine's benefit requires long-term use and appears to be enduring. In support of consistent colchicine use, we found that among colchicine users, both MACE and MIs more commonly occurred during periods when patients had lapsed in their colchicine use compared to periods of active use.

Taken together, the study findings suggest that gout patients with uninterrupted colchicine use may derive more CV benefit than those with inconsistent use, and that colchicine may also exert at least part of its effect at the time of an acute MI. In prior literature, colchicine's proposed mechanism of action in CV disease has been linked to its ability to limit systemic inflammation, reducing the development of atherosclerotic plaque and/or rendering it less prone to rupture [26]. We speculate an additional cardiovascular protective mechanism by which colchicine may act: During an acute plaque rupture or erosion, colchicine may blunt both the ensuing inflammatory cascade and thromboinflammation, thus preventing subsequent vessel thrombosis. Consistent with this possibility, Shah et al. demonstrated both colchicine's ability to inhibit the extent of monocyte–platelet and neutrophil–platelet aggregation and its ability to prevent the development of inflammation during PCI-mediated injury [16,18]. Presuming that colchicine mitigates MI in part by suppressing both acute-onset inflammation and platelet–leukocyte aggregation during and immediately after plaque rupture, its benefit would be most apparent during active use.

The strengths of our study include a large VA population enriched in gout. Our use of a cohort prior to the first (2012) ACR gout treatment guidelines was an advantage since the 2012 guideline recommendations limited the use of extended colchicine and would have made identification of extended colchicine users more difficult. Our use of direct chart identification of gout based on guidelines, in addition to ICD-9 codes (which have already been reported to be suitable for identifying gout patients in VA studies [27]), resulted in the likely exclusion of many gout patients but likely also minimized gout misdiagnoses and increased gout specificity in our study cohort.

Limitations to our study included a relatively small final sample size that limited our power to detect small differences and limited the precision of our findings. The retrospective cohort design resulted in colchicine users and nonusers being imperfectly matched, with differing baseline CV risks and average observation periods, and introduced the possibility of confounding that was not fully addressed with adjusted analyses. Our use of the VA pharmacy database provided reliable data on when and how frequently medications were filled (particularly since veterans receive free or discounted medication and are unlikely to get their medications elsewhere); however, we had no means to guarantee adherence to colchicine prescriptions once filled, and compliance with gout medications is recognized to be suboptimal [28]. This study was also not powered to assess differences in efficacy between dosages of colchicine (though prior trials among non-gout patients suggest that once-daily dosing may be sufficient to reduce cardiovascular events) [7,8]. Additionally, individual patients were exposed to different total dosages of colchicine due to differences in the frequency of colchicine use and changes in dosing instructions over time. CV events were directly confirmed if they occurred in the VA system, and those occurring outside were only captured if documented in VA physician notes. The exclusion of female patients was necessary given the small number of women in our initial cohort (0.6%), but limits the generalizability of our findings to the female population. In light of the limitations inherent to any retrospective study, our findings are hypothesis-generating and prompt the need for further randomized trials.

## 5. Conclusions

In this Veterans Affairs cohort of male gout patients with preexisting coronary artery disease, a cohort of chronic colchicine users had higher CV risk than nonusers, but nonetheless had equivalent rates of MACE. Moreover, patients using colchicine consistently demonstrated lower odds of MACE and may have derived CV protection compared with patients with less consistent use. Consistent colchicine use over the extent of our study period (i.e., years) demonstrated the most appreciable protective benefits, and protection appeared to be increased during periods of active use. This study adds to the growing body of literature supporting the cardiovascular benefit of colchicine in high-risk populations. Prospective randomized trials are necessary to study colchicine's long-term cardiovascular benefit in gout patients, with the potential to impact the standard of care for all gout patients.

**Supplementary Materials:** The following supporting information can be downloaded at: https://www.mdpi.com/article/10.3390/gucdd1010003/s1, Table S1: Comparison of baseline characteristics between all Quartiles (1–4); Table S2: Total Major Adverse Cardiovascular Events and Component Events in Colchicine Users and Nonusers; Table S3: Total Major Adverse Cardiovascular Events and Component Events in Each Colchicine-use Quartile.

**Author Contributions:** Conceptualization, G.H.H. and M.H.P.; methodology, G.H.H., M.H.P. and D.B.C.; software, B.S. and G.H.H.; validation, G.H.H.; formal analysis, G.H.H. and B.S.; investigation, D.B.C. and G.H.H.; resources, D.B.C. and M.H.P.; data curation, G.H.H., M.H.P. and M.T.; writing—original draft preparation, G.H.H.; writing—review and editing, M.H.P., M.T., D.B.C., B.S. and G.H.H.; visualization, G.H.H. and M.H.P.; supervision, M.H.P. and M.T.; project administration, G.H.H.; funding acquisition, N/A. All authors have read and agreed to the published version of the manuscript.

**Funding:** This research received no external funding.

**Institutional Review Board Statement:** This study was approved by the Institutional Review Board of the VA New York Harbor Health Care System, project 1575139. The study was first approved 19 December 2011 and was transitioned according to the 2018 Common Rule on 1 June 2020, such that IRB continuing review is no longer required.

**Informed Consent Statement:** Patient consent was waived with approval of the VA IRB owing to inability to obtain consent from subjects in the setting of a retrospective records-based study.

**Data Availability Statement:** The data presented in this study are available on request from the corresponding author. The data are not publicly available due to privacy, with data including personal identifiers.

**Acknowledgments:** We would like to acknowledge Courtney Pike for coordinating the initial IRB approval and Yuhe Xia for reviewing the statistical analyses.

**Conflicts of Interest:** None of the authors have a conflict of interest regarding this work, but for the purposes of full disclosure, MT reports that he has served as a consultant to Horizon Therapeutics; MHP receives support from the NIH/NCATS (1UL1 TR001445-02), serves/has served as a consultant to Horizon Therapeutics, Swedish Orphan Biovitrum, Federation Bio Inc. and Fortress Bioscience, and has received investigator-initiated research grants (payable to NYU Grossman School of Medicine) from Horizon Therapeutics and Hikma Pharmaceuticals. BS receives grant funding from the NIH NHLBI (1R01HL146206, 3R01HL146206-02S1) and VA Office of Research and Development, serves on the advisory boards for Philips Volcano and Horizons Therapeutics, and serves as a consultant for Terumo medical. DBC is an employee of CymaBay Therapeutics, but this work was related to her prior role as a faculty member at NYU.

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
