# Peer review of "Colchicine Use and Major Adverse Cardiovascular Events in Male Patients with Gout and Established Coronary Artery Disease: A Veterans Affairs Nested Retrospective Cohort Study"

_2813-4583, doi:10.3390/gucdd1010003_

Round 1

Reviewer 1 Report

Manuscript Review:

This is a retrospective cohort study evaluating the association between colchicine use and MACE in patients with gout and CAD in a VA database. The authors reported that patients on colchicine overall had higher baseline risk and compared to non-colchicine users, there was no significant difference in odds of MACE. The manuscript is interesting and is well written. I have some minor comments for the authors’ consideration:

1-Line 90: Can you provide references for the statement that “NSAIDs and glucocorticoids have associations with CV risk”.

2-In Figure 1 legend, please state that NS=non-significant.

3-Figure 4: Please add to the figure legend the statistical method used to assess differences between active and lapse users and to be consistent, if difference in MACE is non-significant, add NS.

Author Response

This is a retrospective cohort study evaluating the association between colchicine use and MACE in patients with gout and CAD in a VA database. The authors reported that patients on colchicine overall had higher baseline risk and compared to non-colchicine users, there was no significant difference in odds of MACE. The manuscript is interesting and is well written.

Response:  We thank the reviewer for their good opinion. Although we saw only a trend in MACE reduction with colchicine use overall, we do note that patients with the most regular colchicine use did have statistically significant lower rates of MACE in our secondary analyses (Part II). We believe that the study is thought-provoking and that it will encourage additional investigations. 

I have some minor comments for the authors’ consideration.

1-Line 90: Can you provide references for the statement that “NSAIDs and glucocorticoids have associations with CV risk”.

Response: Thank you for the suggestion. We now reference the following reports:

  1. Souverein PC, Berard A, Van Staa TP, et al. Use of oral glucocorticoids and risk of cardiovascular and cerebrovascular disease in a population-based case-control study. Heart. 2004;90(8):859-865.
  2. Coxib and traditional NSAID Trialists' (CNT) Collaboration, Bhala N, Emberson J, et al. Vascular and upper gastrointestinal effects of non-steroidal anti-inflammatory drugs: meta-analyses of individual participant data from randomised trials. Lancet. 2013;382(9894):769-779.

Please see page 2, line 92 of our revised manuscript, and appropriate added references (new #23 and 24) on page 15, lines 463-467.

2-In Figure 2 legend, please state that NS=non-significant.

Response: Thank you for pointing this out. We have added this abbreviation explanation to the figure legend. Please see page 7, line 219 of our revised manuscript.

3-Figure 4: Please add to the figure legend the statistical method used to assess differences between active and lapse users and to be consistent, if difference in MACE is non-significant, add NS

Response: We have edited Figure 4 and the associated figure legend to reflect these suggestions.  Please see the revised figure 4 and its legend on page 12, line 278 of our revised manuscript.

Note: Please see the attached revised manuscript to review changes in the body of the text

Reviewer 2 Report

This paper contributes and is important in this area of colchicine usefulness for cardiovascular events in gout patients.

Retrospective analysis of a pharmacy database has limitations and patients adherence to treatment could not be evaluated.

Some points:  Quality of figures (particularly 2a and 3b, could improve.

Do authors have any data about adverse events (gastrointestinal) among colchicine users and if it was a limitation for chronic usage?

Author Response

This paper contributes and is important in this area of colchicine usefulness for cardiovascular events in gout patients.

Response: Thank you.

1-Retrospective analysis of a pharmacy database has limitations and patients' adherence to treatment could not be evaluated.

Response: Thank you for pointing out the limitations of a retrospective analysis utilizing pharmacy database records. In response to this point we now highlight these limitations in the discussion. We are confident, however, that patients of the Veterans Affairs Medical Center filled colchicine almost exclusively at the Veterans Affairs pharmacy and that consistent refills suggests consistent use of medication. Please see page 13, lines 344-348 of our revised manuscript.

2-Some points: Quality of figures (particularly 2a and 3b, could improve.

Response: We do appreciate that the quality of the figures could improve. We have updated the contents of the figures based on feedback from reviewer 3 in addition to improving the design of the figures, including the use of color in all panels. Please see pages 6, 9, 10 and 12 of our revised manuscript for the updated figures. Additionally, we have eps versions of the figures at high resolution in case those are preferred by the journal.

3-Do authors have any data about adverse events (gastrointestinal) among colchicine users and if it was a limitation for chronic usage?

Response: Thank you for raising this important question. While we did not collect data about adverse events associated with colchicine use, literature in both Familial Mediterranean Fever1 and cardiovascular disease2 has demonstrated its safety and tolerability for long-term use, particularly at the once daily dosing of 0.6mg. Of note, in both the COLCOT3 and LoDoCo24 trials, which studied 0.5mg colchicine once daily for approximately 24 months in more than 10,000 subjects, treatment discontinuation and adverse effects, including major gastrointestinal events, were similar among treatment arms. Minor gastrointestinal symptoms (e.g., stomach upset, diarrhea) are more common with colchicine use but tend to resolve with continued use (Nidorf SM, personal communication). In our study, patients who started colchicine and discontinued it early (less than 30 days) or never refilled their script owing to early GI intolerance would have been excluded from the study according to the enrollment design.

In response to the reviewer's comment, we have added a reference to colchicine's tolerability, along with the addition of references to provide more insight into the tolerability of colchicine page 2, lines 63-64, and associated new references (new #11 and 12), page 15, lines 437-440.

References:

1Ben-Chetrit E, Levy M. Colchicine prophylaxis in familial Mediterranean fever: reappraisal after 15 years. Semin Arthritis Rheum. 1991;20(4):241-246.

2Andreis A, Imazio M, Avondo S, et al. Adverse events of colchicine for cardiovascular diseases: a comprehensive meta-analysis of 14 188 patients from 21 randomized controlled trials. J Cardiovasc Med (Hagerstown). 2021;22(8):637-644.

3Tardif JC, Kouz S, Waters DD, et al. Efficacy and Safety of Low-Dose Colchicine after Myocardial Infarction. N Engl J Med. 2019;381(26):2497-2505.

4Nidorf SM, Fiolet ATL, Mosterd A, et al. Colchicine in Patients with Chronic Coronary Disease. N Engl J Med. 2020;383(19):1838-1847.

Note: Please see the attached document below to review changes in the body of the text

Reviewer 3 Report

The authors examine whether the incidence of MACE or CAD in gout is related to the use of colchicine. I have some comments:

The study was censored by the end of 2009. During the last 13 year many changes have taken place, including a more widespread use of colchicine treatment in gout. It would be good to review more recent data in this manuscript.

The diagnosis of gout was in many cases based on the old ARA criteria, and in combination with a GP diagnosis of gout. As the old ARA criteria are not very specific for gout, please report on in how many individuals with gout the different criteria were applied.

The term “cardiovascular risk” is introduced on page 5 but has not been defined or explained before.

In the figures please give numbers of patients, for example in figure 2A. Incidence should be explained as per year or 100 observation years.

Colchicine dosage is an important issue of research in the manuscript, important for part 2. Please describe more in detail how the dosages were calculated, not only in percentage of time, but also the absolute doses per day need to be given, if possible.

How was variation of colchicine dosages handled? Cardiovascular risk is researched and may be related to absolute dosages, fluctuations in dosages, or on-off in colchicine treatment.

Last paragraph page 5. The explanation of the quartiles by % and SD needs to be explained much better.

When odds ratios are calculated, I miss a table comparing all 4 quartiles with colchicine, a description of cases (all and CV outcomes) in the quartiles and information on variables for which adjustments have been made (all 9 factors described for logistic regression analyses?).

Please justify a “lapse period” of only 14 days.

The proposition of an additional protective mechanism is not based on findings in this paper, just speculation, right?

As given by the authors under limitation, the small sample size limits the power to detect differences between colchicine users and non-users as well as the influence of colchicine dosage.

Minor in abstract: the term “minimum >30 continuous days” is not clear. Do you mean <30 days or more than 30 days.

Author Response

The authors examine whether the incidence of MACE or CAD in gout is related to the use of colchicine. I have some comments:

1-The study was censored by the end of 2009. During the last 13 year many changes have taken place, including a more widespread use of colchicine treatment in gout. It would be good to review more recent data in this manuscript.

Response: Thank you for raising this issue. We respectfully disagree with the reviewer's assertion that there is currently a more widespread use of colchicine treatment in gout and are not aware of any specific data to that effect.

We chose specifically not to analyze later data for the following reason. Prior to 2012, the extended use of colchicine was commonly employed by both primary care providers and rheumatologists as an adjunct or alternative to urate lowering therapy for the treatment of gout. A recommendation in 1971 stated that “treatment with one tablet thrice daily is often effective during intercritical periods to prevent articular flare-ups.”5 In 1988, Wallace and Singer recommended that “acute gouty recurrences can be prevented…with colchicine and without the use of anti-hyperuricemia drugs.”6 Even when used as prophylaxis during urate lowering, the duration of therapy was quite varied. For example, one report in 1996 recommended “continuation of [colchicine] therapy for at least a year after the serum urate concentration has returned to a normal level.”7 Equally ambiguously, the 2006 EULAR gout treatment guidelines recommend colchicine for flare prophylaxis for the “first months” of urate lowering therapy, but did not specifically address when to discontinue colchicine.8 Thus, colchicine treatment in those early days was characterized by widely variable approaches to its use. In contrast, the 2012 ACR guidelines for the management of gout specified a duration of 6 months of colchicine therapy for gout prophylaxis when starting urate lowering treatment, for a few days during flares, and no recommendation for colchicine use as an independent, stand-alone and/or persistent anti-inflammatory9. The 2020 guideline was even shorter, specifying “3 to 6 months” during initiation of urate lowering therapy for any patient.10 Thus, these guidelines changed the standard of practice in the rheumatology community and limited extended colchicine use, with or without urate lowering, which would have been an unacceptable set of conditions for our study and would not have provided us with useful information.

Since we sought to study a group of patients whose extent of colchicine use was varied, and included individuals who were on colchicine for shorter and longer periods, as well as those who might be taking colchicine independently of allopurinol, we concluded that any data after 2012 would be less relevant to our study. For these purposes, therefore, we chose to use of data until the end of 2009 (we could have extended to 2012 but opted for a 10-year window from the start of 2000 to the end of 2009). Our approach was borne out, since both the time spent on colchicine (months to years) and the percent of observation time (less than 10% to more than 90%) on colchicine varied widely among our patient population, providing us with useful data. Although not discussed extensively in the manuscript, the percentage of patients on colchicine for greater than 30 days without allopurinol or other urate lowering therapy was substantial during this period (in fact use of colchicine was more frequent without than with allopurinol use, as shown in Table 1). This would not have been true in the later period since the guidelines make no recommendation for colchicine alone.

We thank the reviewer and recognize that we did not make this issue clear in our prior version. In response to the reviewer’s comment, we now have provided additional text in the manuscript to explain the importance of using data pre-2012 to retrospectively study colchicine’s impact. Please see our revised manuscript, page 3, lines 104-106.

References:

5Goldinger SE. Treatment of gout. NEJM 1971;285(23):1306.

6Wallace SL And Singer JZ. Therapy in gout. Rheumatic Disease Clinics of North America 1988;14(2):441-457.

7Emmerson BT. The management of gout. NEJM 1996;334:445-451.

8Zhang W, Doherty M, Bardin T, et al. EULAR evidence-based recommendations for gout. Part II: Management. Report of a task force of the EULAR Standing Committee for International Clinical Studies Including Therapeutics (ESCISIT). Ann Rheum Dis. 2006;65(10):1312-1324.

9Khanna D, FitzGerald JD, Khanna PP et al. 2012 American College of Rheumatology guidelines for the management of gout. Part 2: therapy and antiinflammatory prophylaxis of acute gouty arthritis. Arthritis Care Res 2012;64(10):1447-61.

10FitzGerald JD, Dalbeth N MIkuls T, et al. 2020 American College of Rheumatology Guideline for the management of gout. Arthritis Rheum 2020;72(6):879-95.

2-The diagnosis of gout was in many cases based on the old ARA criteria, and in combination with a GP diagnosis of gout. As the old ARA criteria are not very specific for gout, please report on in how many individuals with gout the different criteria were applied.

Response: We first would comment that we did not rely on the ARA criteria alone. In order to be reviewed for inclusion, patients first needed to have an ICD-9 code for gout or hyperuricemia. This is particularly relevant, since Singh published in Arthritis Research & Therapy that ICD-9 codes alone were for sensitive and specific for the identification of gout in a VA population (and our study was limited to veterans in the VA system)11. Our study screening goes one step further and applies the ARA criteria to those already identified using ICD-9 codes, thus providing improved specificity to our cohort. Of the 7,819 patients screened using ICD-9 codes (for gout or hyperuricemia, but the majority for gout), 1,638 (20.9%) were ultimately categorized as having gout using our strict study criteria (Figure 1).

Regarding our use of ARA criteria12, we first would point out that the specificity for the ARA criteria, in one study comparing the specificity of a number of different gout criteria, was 84.5%; and that in that study the ARA criteria were the most specific available at the time.13 Combined with the use of ICD-9 codes, we feel confident that these criteria ensured that the population under study was specific for gout. In addition, we used the ARA criteria for two pragmatic reasons: 

First, this dataset (used in papers previously published by our group, as referenced in the current manuscript) was first developed in 2013-2014, after the release of the 2012 ACR gout treatment guidelines but before the newer 2015 ACR/EULAR gout classification criteria; therefore the new ACR/EULAR criteria were not applied for the identification of gout at the time of cohort development since these criteria did not yet exist.

Second, we considered re-analyzing our data by applying the 2015 ACR/EULAR classification criteria but discovered that, even with a rigorous chart-by-chart review, the 2015 criteria were really designed for the prospective enrollment of gout patients into clinical trials and were poorly suited for use in retrospective cohorts for multiple reasons. Primarily, the 2015 criteria rely on many aspects of the history that are not always recorded in the medical record, such as time to maximum intensity of pain, time to resolution of flare, and the specific character of the pain. Secondly, the 2015 criteria rely on imaging modalities that were not widely available pre-2012 when our patients' data were recorded (such as DECT and musculoskeletal ultrasound). Thus we found the ARA criteria were more suited for data extraction from the medical record. In applying the ARA criteria, each patient chart was carefully reviewed according to a verified extraction algorithm. Since even the ARA criteria had limitations during retrospective review, we were forced to apply modifications of those criteria in order to complete our analyses. These included accepting urate-lowering therapy as a surrogate for hyperuricemia (urate levels not always available in the laboratory record), and accepting GP diagnosis of gout as a surrogate for some other criteria (but not independent of those criteria). While many patients who really had gout were clearly left out of our analysis for not having sufficient data to meet ARA criteria, we are extremely confident that those who were included unequivocally had gout.

In response to the reviewer’s query, we now provide clarification on the specificity of inclusion of subjects based on the dual application of ICD-9 and ARA criteria. For the reviewer's information, here are the data on which criteria were used to identify patients for inclusion:130 patients met inclusion criteria by a crystal diagnosis, 681 were diagnosed with gout by a rheumatologist (and were not further reviewed), 223 met ≥ 6/12 ARA classification criteria (the ARA minimum for gout diagnosis), and 604 had ≥ 4/12 ARA criteria and additionally had a primary care diagnosis of gout in the medical record. We now add this information to our revised manuscript; please see page 4, lines 171-174.

11Singh JA. Veterans Affairs databases are accurate for gout-related health care utilization: a validation study. Arthritis Res Ther. 2013;15(6):R224. doi:10.1186/ar4425

12Wallace SL, Robinson H, Masi AT, et al. Preliminary criteria for the classification of the acute arthritis of primary gout.  Arthritis Rheum 1977;20(3):895-900.

13Jatuworapruk K, Lhakum P, Pattamapaspong N, et al.Performance of the existing classification criteria for gout in Thai patients presenting with acute arthritis. Medicine (Baltimore) 2016;95(5):e2730

3-The term “cardiovascular risk” is introduced on page 5 but has not been defined or explained before.

Response: Thank you for the suggestion. The term "cardiovascular risk" is now defined on page 5 as an increased prevalence of prior myocardial infarction, prior PCI, NSAID use, and a higher baseline LDL Cholesterol. Please see our revised manuscript, page 6, lines 200-201.

4-In the figures please give numbers of patients, for example in figure 2A. Incidence should be explained as per year or 100 observation years.

Response: We appreciate your point that the different observations periods should be taken into account when calculating the incidence of MACE; we now reported the incidence of MACE per year of observation in all sub-sections and have remade our figures to reflect this. The incidence of MACE in colchicine users per year were observed to be 2.9%, while the comparative incidence in non-users was 3.7% (p=0.38). The incidence of MACE per year in Quartile 4 versus Quartiles 1-3 was 1.5% and 3.4% respectively (p=0.01) As a result of following this suggestion, our figures now reflect rates and therefore it is not relevant to include numbers of patients in the figures themselves. Please see page 4, lines 157-159 of our revised manuscript, and our revised Figures 2A and 3B.

In response to the reviewer’s request for numbers of patients, we have created Supplemental Tables 2 and 3, documenting the absolute number of events in each subgroup. Supplemental Table 2 compares the numbers for unadjusted MACE between colchicine users and non-users, and Supplemental Table 3 compares the numbers for unadjusted MACE in each individual quartile.

5-Colchicine dosage is an important issue of research in the manuscript, important for part 2. Please describe more in detail how the dosages were calculated, not only in percentage of time, but also the absolute doses per day need to be given, if possible.

Response: Using the medication fill records from the VA pharmacy database, daily dosages of all colchicine prescriptions were collected for each patient and averaged across that individual’s colchicine use period. We then calculated the mean daily dosages of colchicine used within each sub-group, which is reported in Tables 1 and 2. The vast majority of colchicine users were prescribed colchicine at least 0.6 mg once daily (90.3%). Most colchicine users were prescribed once daily dosing at 0.6mg (58.8%), others were prescribed twice daily dosing at 0.6mg (13%) or alternated between daily and BID dosing during their observation periods (18.5%). A small number of patients were taking lower doses (9.7%), typically as a renal adjustment. The breakdown of dosages is now reported in our revised text on page 5, lines 190-193.

6-How was variation of colchicine dosages handled? Cardiovascular risk is researched and may be related to absolute dosages, fluctuations in dosages, or on-off in colchicine treatment.

Response: Thank you for this important question.

In terms of the absolute dose per day needed to be given for benefit, we found no difference in effect between 0.6 mgs daily or BID, however our numbers were not large enough for confident analysis (not shown). However, we note that 0.5 mg daily is the dose at which the cardioprotective effects of colchicine were demonstrated in the LoDoCo14, LoDoCo24 and COLCOT3 trials (more than 10,000 subjects not specifically with gout), suggesting that the lower dose is likely to be sufficient. Modeling efforts also suggest that 0.5 mgs daily should be sufficient in most patients to achieve the presumed therapeutic blood level.15 Of course, the only way to guarantee consistent dosages and assess difference would be through rigorous prospective clinical trials. In response to the reviewer in this regard, we now include content in our discussion addressing the limitations of the study in distinguishing between doses. Please see Discussion, page 13, lines 348-351 in our revised manuscript.

Regarding fluctuations in dosages, the large majority of our subjects (81.5%) maintained the same dose of colchicine even when they went off and then returned to treatment. Therefore our study does not permit us to address the impact of dose fluctuations. To be truly meaningful, studies of colchicine dose would also need to account for serum levels which may vary from patient to patient. Since there is no FDA approved, commercially available peak and trough colchicine blood test, there is no information available for us to study this question, even if dosing changes had been common. We now acknowledge this as a limitation in our Discussion. Please see page 13, lines 351-353 of our revised manuscript.

Regarding the issue of “on-off” in colchicine use, this is exactly what our exploratory analysis of active use colchicine use versus lapse addressed. The data suggest that colchicine is more effective when used consistently, and more effective during “on” periods than during “off” periods. Please see Figure 4 and related text.

14Nidorf SM, Eikelboom JW, Budgeon CA, et al. Low-dose colchicine for a secondary prevention of cardiovascular dissease. J Am Coll Cardiol 2103;61(4):404-410.

4Nidorf SM, Fiolet ATL, Mosterd A, et al. Colchicine in patients with chronic coronary disease.  NEJM 2020;383(19):1838-1847.

3Tardif JC, Kouz S, Waters CC, et al.  Efficacy and safety of low-dose colchicine after myocardial infarction. NEJM 2019;381(26):2497-2505.

15Karatza E, Ismailos G, Karalis V. Colchicine for the treatment of COVID-19 patients: efficacy, safety, and model informed dosage regimens. Xenobiotica 2021:51(6):643-656.

7-Last paragraph page 5. The explanation of the quartiles by % and SD needs to be explained much better.

Response: Thank you for the opportunity to clarify this. We have added a brief explanation of our methods for calculating the “consistency in colchicine use” and how to interpret these numbers. Please see page 7 of our revised manuscript, lines 227-231.

8-When odds ratios are calculated, I miss a table comparing all 4 quartiles with colchicine, a description of cases (all and CV outcomes) in the quartiles and information on variables for which adjustments have been made (all 9 factors described for logistic regression analyses?).

Response: We did have this data but did not include it in the manuscript due to length restrictions. To address the reviewer’s request, we now provide two supplemental tables: 1) comparing baseline characteristics between all quartiles and 2) comparing MACE events between each individual quartile. The samples sizes of each individual quartile would be too small to calculate meaningful p-values. Please see Supplemental Tables 1 and 3.

All 9 factors were adjusted for in the reported odds ratio comparing MACE in Q4 vs. Q1-3, but clearly we did not make this explicit and now do so. Please see page 8 line 243 of our revised manuscript.

9-Please justify a “lapse period” of only 14 days.

Response: We apologize for the confusion. We did not define a lapse period of only 14 days. Rather we defined that any lapse period would only be considered to start 14 days after the end of the last prescription. In effect, a lapse period was defined as starting when a patient had completed their colchicine prescription and had adequate time to metabolize and excrete the drug of interest (94-97% elimination at 5 half-lives, which is about 5-7 days for colchicine). 14 days was a chosen time frame that was somewhat longer than 5 half-lives to also account for the possibility of slow metabolizers, and to account for any extra/residual pills that patients might have had in their bottle and kept taking after they were predicted to have run out. Thus, we feel confident that any cardiovascular event occurring during a lapse period would have occurred without the, presumed cardiovascular protective, effect of colchicine (off colchicine).

In response to the reviewer, we now clarify our definition of lapse and provide a rationale for the 14-day delay in starting the lapse period. Please see page 11, lines 268-271 in our revised manuscript.

10-The proposition of an additional protective mechanism is not based on findings in this paper, just speculation, right?

Response: Essentially, right. Having identified a potential benefit for colchicine, we believe that the reader might wish to have insight into how such a benefit might occur. The proposed mechanism we discuss is supported by physiologic research by Shah et al., in “Effect of Colchicine on Platelet-Platelet and Platelet-Leukocyte Interactions: a Pilot Study in Healthy Subjects”18, and is further supported by our findings that 1) events occurred numerically more often when colchicine use was lapsed as compared to periods of active use, and 2) more consistent colchicine is associated with a lower odds of MACE. Taken all together, colchicine’s protective cardiovascular effect is hypothesized to be most evident at the time when therapeutic levels are achieved.

In response to the reviewer's question, we have substituted the word "speculate" rather than "propose" to make clear that the study is consistent with, but in no way demonstrates, this possible mechanism.  please see page 13, line 318 of our revised manuscript.

18 Shah B, Allen N, Harchandani B, et al. Effect of colchicine on platelet-platelet and platelet-leukocyte interactions: a pilot study in healthy subjects. Inflammation 2016;39(1):182-189.

11-As given by the authors under limitation, the small sample size limits the power to detect differences between colchicine users and non-users as well as the influence of colchicine dosage.

Response: We recognize that this represents a limitation to our study; however, our small sample size (hundreds) also allowed us to perform chart-by-chart evaluations of our patient records, which provided some compensatory strengths. Overall, we believe our findings prompt the academic community to further consider the role of colchicine in gout and CAD, and highlights the need for prospective randomized trials.

In response to the reviewer’s comment, we now expand our mention of sample size as a limitation.  Please see page 13, lines 357-358 of our revised manuscript.

12-Minor in abstract: the term “minimum >30 continuous days” is not clear. Do you mean <30 days or more than 30 days.

Response: Thank you for pointing out the redundancy in this phrase. We have corrected this in the abstract. Please see page 1, lines 19-20 of our revised manuscript.

Note: Please see the attached revised manuscript to view changes within the body of text

Round 2

Reviewer 3 Report

The authors have thoroughly dealt with the reviewers' comments, and I agree to a high degree with their reasoning and to their alterations done in the manuscript. No further comments.